# A Comparative Study of Skin Changes in Different Species of Mice in Chronic Photoaging Models

**DOI:** 10.3390/ijms241310812

**Published:** 2023-06-28

**Authors:** Meifen Lin, Xiaoran Liu, Xueer Wang, Yinyan Chen, Yijia Zhang, Jinfu Xu, Lingwei Bu, Yarui Zhang, Fengting Liang, Xinyue Zhang, Bingli Huang, Min Zhang, Lin Zhang

**Affiliations:** 1GDMPA Key Laboratory of Key Technologies for Cosmetics Safety and Efficacy Evaluation, NMPA Key Laboratory for Safety Evaluation of Cosmetics, Department of Histology and Embryology, School of Basic Medical Sciences, Southern Medical University, Guangzhou 510515, China; linmf8995@163.com (M.L.); wangxueer123@smu.edu.cn (X.W.); xujinfu@126.com (J.X.); zhangyruia@163.com (Y.Z.);; 2Guangzhou Dublin International College of Life Sciences and Technology, South China Agricultural University, Guangzhou 510642, China; 3School of Public Health, Southern Medical University, Guangzhou 510515, China

**Keywords:** chronic skin photoaging model, C57BL/6J mice, superoxide dismutase, malondialdehyde, ultraviolet

## Abstract

This study aimed to design a novel mouse model of chronic photoaging. We used three different species of mice (C57BL/6J, ICR, and KM) to create a chronic photoaging model of the skin. The irradiation time was gradually increased for 40 consecutive days. The skins of the mice were removed on day 41 and subjected to staining to observe them for morphological changes. Immunohistochemistry was used to detect tumor necrosis factor-α (TNF-α) and p53 expression; superoxide dismutase (SOD) and malondialdehyde (MDA) were measured as well. Compared with C57BL/J mice, which showed hyperpigmentation, the irradiated skin of ICR and KM mice showed more obvious skin thickening and photoaging changes of the collagen and elastic fibers. KM mice had higher levels of inflammation, oxidative stress, and senescent cells. Compared with the 5-month-old KM mice, the photoaging changes of the 9-month-old KM mice were more pronounced, the SOD values were lower, and the MDA values were higher. In summary, KM mice have higher levels of abnormal elastic fibers, inflammation, cellular senescence, and oxidative stress than ICR mice, and are more suitable for studies related to chronic skin photoaging. C57BL/6J mice were found to be suitable for studies related to skin pigmentation due to photoaging.

## 1. Introduction

Skin aging can be classified into endogenous and exogenous aging [1]. Endogenous aging is the process of decline and degradation of the whole organism, and is influenced by both endocrine and genetic factors. Exogenous aging occurs gradually from childhood, and is caused by long-term exposure to sunlight, pollution, ionizing radiation, and toxins. Among these, ultraviolet (UV) light is the most important factor in exogenous skin aging [1], which is called skin photoaging when caused by UV from sunlight. Acute UV irradiation can cause skin damage, which manifests as sunburns, peeling, and inflammation. Chronic UV irradiation can cause aging in skin cells and tissue structure. UV can be classified by wavelength as UVA (400–315 nm), UVB (315–280 nm), and UVC (280–100 nm). UVB is mostly absorbed by the epidermis, and causes damage to cellular DNA [2,3,4,5,6]. UVA can penetrate deep into the epidermis and induce various types of DNA lesions through direct and indirect mechanisms, and is now considered to be the main factor in skin photoaging [7,8,9]. Both UVA and UVB are important factors in photoaging, and the use of combined UVA and UVB irradiation is significant for photoaging models.

The classical mechanisms of photoaging include cellular senescence, inflammatory responses, and oxidative stress [1]. Cellular senescence is an important driver of photoaging [7]; p53 signaling is involved in the systemic regulation of the aging process and plays an important regulatory role [10]. UV light induces the production of cytokines such as tumor necrosis factor-α (TNF-α) in epidermal keratinocytes and dermal fibroblasts. TNF-α stimulates the release of other cytokines, chemokines, and adhesion molecules, which trigger skin inflammation [11], making TNF-α an important indicator of inflammation. There is a close relationship between oxidative stress and inflammatory response [12,13], and oxidative stress is considered an important factor contributing to aging [14].

Recent studies have found that skin aging and various chronic diseases are related [15,16]. Animal models of chronic skin photoaging have core value in the study of photoaging. The selection of an appropriate skin photoaging model can help reveal the molecular mechanism of skin photoaging, and can be used to conduct effective prevention and treatment intervention research. ICR and KM mice have been utilized in chronic photoaging models [17,18,19,20], while C57BL/6J mice have been utilized in acute photoaging models [21]. At present, there are no comparative studies on chronic skin photoaging models using different strains of mice. Therefore, in this study 5-month-old (5M) C57BL/6J, 5M ICR, and 5M KM mice and 9-month-old (9M) KM mice were used as chronic photoaging models. First, we aimed to investigate the effect of species on chronic photoaging in mice and compare the changes in skin appearance, morphology, inflammatory response, senescent cells, oxidative stress levels, and other aspects of the three groups of mice of the same age were compared to determine which mice are most suitable as mouse models for chronic photoaging. Then, we aimed to investigate the effect of age on chronic photoaging in mice and to provide a suitable mouse model for the study and treatment of skin photoaging.

## 2. Results

### 2.1. Chronic Photoaging Model of Mouse Skin

The groupings of the mice are shown in Figure 1a. The design is shown in Figure 1b, where the upper back skin of each mouse was the self-control (unirradiated site) and the lower back was the UV irradiation site. For the self-control, a mouse fixator was designed for this study (Figure 1c), and the mice were fixed without anesthesia. During the photoaging molding process of the mouse skin, the amount of UVA and UVB radiation was detected using UVA and UVB detection instruments (Figure 1d), and the effect of the tin foil covering the UV light was detected. Mouse skin photoaging modeling was performed daily (Figure 1e).

### 2.2. General Observations in Chronic Photoaged Skin of Mice

The back skin of the mice at days 0, 5, 10, 20, 30, and 40 was recorded using a camera and dermoscope to observe the skin changes during the process of chronic photoaging. The unirradiated part of the three different species of mouse was smooth, with no obvious redness or desquamation (Figure 2a–c). The irradiated skin of the 5-month-old ICR mice showed typical characteristics of chronic photoaged skin, such as obvious thickening, leathery skin, redness, and desquamation. The irradiated skin of the 5-month-old KM mice showed more obvious photoaging features, such as thickening, leathery skin, swelling, and desquamation. The irradiated skin of the C57BL/6J mice showed desquamation on day 5, and mainly showed pigmentation on days 10–40; other photoaging characteristics were not obvious. The results of the general skin observations suggested that 5-month-old ICR and KM mice are suitable for studies on epidermal morphological changes in chronic photoaging skin, while C57BL/6J mice are suitable for studies on changes related to chronic photoaging pigmentation. To investigate whether the skin of KM mice of different ages showed differences in the process of chronic photoaging, the skin of 9-month-old KM mice was included (Figure 2d). It was found that skin photoaging in the irradiation site of 9-month-old KM mice was more obvious than that of 5-month-old KM mice.

### 2.3. Skin-Related Physiological Changes in Chronic Photoaged Mice

Compared with the unirradiated skin, the percutaneous water loss (TEWL) of the irradiated skin of all mice was significantly increased, suggesting significant skin barrier damage in all chronic photoaged mice (Figure 3a). Among 5-month-old mice, the chronic photoaged skin of ICR mice showed the largest increase in TEWL, followed by KM mice, with the smallest change observed in C57BL/6J mice. The increase in TEWL in 9-month-old KM mice was greater than that in 5-month-old KM mice. The measurement of skin epidermal moisture content of mice using physiological instruments showed that the moisture content of all mice exposed to UV irradiation was lower than that of the unirradiated skin (Figure 3b). Among 5-month-old mice, C57BL/6J mice had the largest decrease in skin moisture content, followed by KM mice, with the smallest change observed in ICR mice. The decrease in moisture content in the 9-month-old KM mice was greater than that in the 5-month-old KM mice, indicating that the water retention capacity of the skin epidermis of mice with chronic photoaging was decreased, leading to skin dryness.

Skin elasticity measurements showed that the skin elasticity at the UV irradiation site of all mice was lower than that of the unirradiated skin (Figure 3c). Among 5-month-old mice, KM mice showed the largest reduction in skin elasticity, followed by ICR mice, with the smallest change observed in C57BL/6J mice. The decrease in skin elasticity observed in the 9-month-old KM mice was less than that in the 5-month-old KM mice. Further skin ultrasound results showed a decrease in dermal density of all irradiated skin (Figure 3d,e). Among 5-month-old mice, KM mice showed the largest decrease in dermal density, followed by ICR mice, with the smallest change observed in C57BL/6J mice. There was no difference in density reduction of KM mice at different months.

### 2.4. Histological Changes in the Skin of Chronic Photoaged Mice

Hematoxylin and eosin (H&E) staining (Figure 4a–d) revealed that the unirradiated skin of all mice showed normal skin thickness, complete structure, orderly distribution, and normal arrangement of dermal fibers. Irregularly thickened skin epidermis was observed at all irradiated sites in the mice. Among all mice, 5-month-old ICR and KM mice showed significant epidermal thickening, whereas the epidermis thickened slightly in 9-month-old C57BL/6J mice and 5-month-old KM mice. Masson staining further showed that dermal collagen fibers were disordered at all irradiation sites and that collagen fiber bundles became thinner, curled, and broken. Among 5-month-old mice, ICR and KM mice showed the most obvious changes. Compared to 5-month-old KM mice, the changes to the dermal collagen fibers in 9-month-old KM mice were more obvious. Gomori staining was used to observe the changes in the elastic fibers of the mouse dermis. The elastic fibers of the skin at the irradiated sites of all the mice were thickened, broken, twisted, and deformed, and some of them were clumped up. The changes in dermal elastic fibers in KM mice were the most obvious in 5-month-old mice, followed by ICR mice. The changes in the dermal elastic fibers of 9-month-old KM mice were similar to those of 5-month-old KM mice.

### 2.5. Increased Expression of TNF- α and p53 in the Skin of Chronic Photoaged Mice

We selected TNF-α as an indicator of inflammation and p53 as an indicator of aging to compare the responses of three different species of mice to inflammation and aging (Figure 5). As shown by immunohistochemical staining of TNF-α (Figure 5a,c), compared with the unirradiated skin, the expression of TNF-α and the percentage of positive area of TNF-α expression sites in the epidermis and dermis were significantly increased in the irradiated skin of all mice. Among the 5-month-old mice, the expression of TNF-α was the most significant in KM mice, followed by ICR mice and C57BL/6J mice. TNF-α expression was significantly higher in 9-month-old KM mice than in 5-month-old KM mice. These results indicate that a skin inflammatory response was observed in all strains of mice during chronic photoaging, most prominently in KM mice.

Immunohistochemical staining of p53 showed that, in comparison with the unirradiated skin, the number of p53 positive cells in the irradiated skin of all mice was significantly increased, and were mainly concentrated in the epidermis, indicating that in the process of chronic photoaging all strains of mice showed senescence of the skin epidermal cells (Figure 5b,c). Among the mice, KM mice had the largest number of p53 positive cells for both 5- and 9-month-old mice, whereas ICR mice had the smallest number of p53 positive cells.

### 2.6. Increased Levels of Oxidative Stress in the Skin of Chronic Photoaged Mice

Oxidative stress is one of the mechanisms that underlie chronic photoaging [22]. Superoxide dismutase (SOD) is an important antioxidant enzyme that removes superoxide anion free radicals, which can protect cells from damage caused by oxygen free radicals. Malondialdehyde (MDA) is an end product formed by the reaction between lipids and oxygen free radicals. UV radiation can induce the production of a large amount of reactive oxygen species, wherein SOD is greatly consumed and MDA accumulates; therefore, the oxidative stress process can be evaluated via SOD and MDA. After the chronic photoaging process, the level of SOD was significantly decreased after UV irradiation, with the most obvious change in 5-month-old KM mice, followed by 5-month-old C57BL/6J mice, while the decrease in SOD levels in 5-month-old ICR mice was not significant (Figure 6a). The level of MDA (Figure 6b) increased after UV irradiation, with the most obvious change in 5-month-old KM mice, followed by 5-month-old ICR mice, and the smallest change found in C57BL/6J mice. In KM mice of different ages, the oxidative stress level of 5-month-old KM mice was more obvious. The basal SOD value of 9-month-old mice was lower than that of 5-month-old mice, while the basal MDA value was higher than that of 5-month-old mice, indicating that oxidative stress levels are affected by natural aging.

## 3. Discussion

Changes in human skin occur with age, and repeated exposure to external stimuli such as the sun accelerates skin photoaging [1,23], especially with the serious destruction of the atmospheric ozone layer. The selection of suitable animal models and modeling methods is helpful for exploring the molecular mechanisms of skin photoaging and prevention of skin photoaging. This study aimed to identify a rapid and effective modeling method for chronic photoaging modeling and to provide a reference for the selection of mouse species. At present, there are deficiencies in the photoaging modeling process. First, the irradiation period for chronic photoaging modeling is long (12 weeks in a study by Li et al. [19]), with the current shortest modeling period proposed by Zheng et al. [24] at 8 weeks; moreover, there remains the problem of non-continuous irradiation. Second, UV irradiation equipment is expensive or lacking, the irradiation equipment used is not specifically designed for modeling of animal photoaging [19,21,24], and unirradiated parts are not guaranteed to be protected from UV injury during the irradiation process. Meanwhile, the problem of animal fixation during irradiation has not been solved [25]. To solve these problems, the present study seeks to improve the modeling method of chronic photoaging mice by designing a UV irradiation box for photoaging modeling (including adjustable heights for UVA and UVB light sources) and a matching mouse fixator which can accommodate multiple mice simultaneously. Mouse fixation can protect the non-irradiated parts from UV damage and does not require anesthesia, greatly reducing the harm caused by administration of anesthesia to mice in line with animal ethical requirements. In the UV irradiation cycle, this study adopted the method of continuous irradiation for 40 days, gradually increasing the daily irradiation time to 30 min, which shortened the irradiation cycle while ensuring consistent irradiation intensity. Throughout the experiment, the mice did not have any serious UV damage such as skin breaking or bleeding. After UV irradiation, the responses of the three strains of mice in terms of skin appearance, tissue morphology, inflammatory response, cellular senescence, and oxidative stress responses were consistent with the characteristics of chronic photoaging reported by Wang et al. [25], such as rough and leathery appearance, peeling, epidermal thickening, dermis thinning, collagen fiber degradation disorder, elastic fiber degeneration and clumping, skin inflammation, and oxidative stress response. The photoaging response was similar to that of human photoaging, which demonstrates that this experiment successfully established a novel, rapid, operable, and low-cost method for chronic photoaging in animals that is applicable in experimental research on photoaging.

This study further compared the morphological and biological changes in C57BL/6J, ICR, and KM mice aged five months during chronic photoaging. As found in the present study, C57BL/6J mice showed significant skin pigmentation after UV irradiation, while compared with ICR mice and KM mice, the skin changes of C57BL/6J mice in the process of chronic photoaging, such as skin morphology, collagenous fiber, elastic fibers, inflammatory reaction, apoptosis, and oxidative stress, were the most atypical changes. These results are consistent with the findings of melanin protecting UV-irradiated skin [1,26], suggesting that C57BL/6J mice are suitable for the study of skin pigmentation induced by photoaging but are not suitable for use in chronic photoaging models, as their black hair interferes with the observation of gross skin appearance. ICR and KM mice are albino mice with no melanin in the skin, and their hair color does not affect the observation of skin changes. Hence, it is more suitable to choose ICR or KM mice to continuously observe changes in gross skin appearance in the process of chronic photoaging. In this study, ICR mice showed similar changes in general appearance, tissue morphology, and collagen fibers to KM mice, and both are similar to the chronic photoaging characteristics in humans [2,23]. However, KM mice showed better changes in elastic fibers, inflammation, cellular aging, and oxidative stress than ICR mice. and had more typical characteristics of chronic photoaging. Therefore, KM mice are more suitable for use in the study of chronic photoaging.

There is a chronological aging process in which all functions of the skin decline with age [2]. Studies have shown that 5-month-old mice are equivalent to 20- to 30-year-old humans, while 9-month-old mice are equivalent to 50- to 60-year-old humans [27]. Compared to 5-month-old KM mice, the normal skin epidermis of the 9-month-old KM mice had uneven skin thickness, slender collagen fiber bundles, reduced elastic fibers, and decreased skin water retention ability, barrier repair ability, and elasticity, which appear to be characteristic of chronological aging [28] and are similar to the chronological aging characteristics of humans [2]. Compared with 5-month-old KM mice, 9-month-old KM mice showed similar changes in collagen degradation and inflammation; however, the changes in SOD and MDA in the oxidative stress response were not obvious, which is not characteristic of typical chronic photoaging. Therefore, using 5-month-old mice can reduce interference from the natural aging process to a certain extent, and 9-month-old mice can be used in studies of the effects of natural aging in combination with photoaging.

## 4. Materials and Methods

Specific pathogen free (SPF) mice (three C57BL/6J, three ICR, and three KM mice, all 5 months old) were used to compare skin changes in a chronic photoaging model. Three SPF KM mice, (9 months old, female) were used to compare skin changes in KM mice at different months during the chronic photoaging model. The mice were provided by the Laboratory Animal Center of Southern Medical University and fed standard animal feed. All mice were raised according to the Guidelines for Animal Experimentation of the Laboratory Animal Center of Southern Medical University, and the study was approved by the Experimentation Animal Ethics Committee of Southern Medical University (protocol code SMUL2023054, 14 April 2023). The UV irradiation device was a self-made UV irradiation box equipped with three 60 cm UVA (340 nm, 40 W) and one 60 cm UVB (313 nm, 40 W) tubes, which could stably provide UVA at 1 mW/cm^2^ and UVB at 0.15 mW/cm^2^ intensity. A laboratory special mouse fixator, which could isolate the UV light with a hollowed-out part (21 × 13 mm) that exposed the back skin of the mouse, was used. The UVA and UVB intensity detectors were purchased from Shanghai Sigma High Technology Co., Ltd., Shanghai, China. The dermatoscope (Trichoscan HD) was purchased from Dermoscan, Regensburg, Germany. The percutaneous water loss instrument (AF200) was purchased from Biox, London, UK, and the skin polyguide physiological instrument and skin composition structural analysis system (DUB-MSTC) were purchased from TPM, Germany. The paraffin embedding, paraffin slicing machine, and Leica forward microscope were purchased from Leica, Wetzlar, Germany. The H&E and Masson three-color staining kits were purchased from Fuzhou Maixin Biotechnology Development Co., Ltd., Fuzhou, China, and the Gomori elastic fiber staining solution was purchased from Beijing Solaibao Technology Co., Ltd., Beijing, China. Primary antibody p53 was purchased from Thermo Fisher, Waltham, MA, USA. Primary antibody TNF-α was purchased from Proteintech, Rosemont, IL, USA. Goat anti-rabbit IgG/HRP polymer and DAB Kit were purchased from Beijing Zhongshan Jinqiao Biotechnology Co., Ltd., Beijing, China. Western and PI cell lysates were purchased from Beyotime Biotech. Inc., Shanghai, China. Total SOD colorimetric cartridge (WST-1) and MDA colorimetric cartridge (TBA) were purchased from Wuhan Elabscience Biotechnology Co.,Ltd, Wuhan, China.

The mice were fed adaptively for one week after purchase. After the adaptation period, three mice from each group were randomly selected and divided into 5-month-old C57BL/6J, 5-month-old ICR, 5-month-old KM, and 9-month-old KM groups. The day before UV exposure began, the back hair of the mice was removed using a mild hair removal cream. In the course of 40 consecutive days of irradiation, hair removal was performed according to the hair growth of the mice, and the frequency was once every 5–7 days. In order to avoid any influence of hair removal on ultraviolet irradiation, we would irradiate the mice 12 h after hair removal. The lower back of the mice was the UV-irradiated part (the exposed skin area was approximately 0.21 × 0.13 cm), the upper back was the non-irradiated part, and the photographing position was marked with a marker pen.

The mice were fixed with a special mouse fixator; the front part of the fixator (non-irradiated skin) was wrapped with aluminum foil to fully block the UV light and was placed in a self-made UV irradiation box. The chronic photoaging molding period was 40 days, and the irradiation doses were 1 and 0.15 mW/cm^2^ for UVA and UVB, respectively, wherein the duration was 10 min on days 1–5, 20 min on days 6–10, and 30 min on days 11–40 (Figure 1).

After 40 consecutive days of irradiation, the mice were anesthetized using intraperitoneal injection of 2% pentobarbital sodium and the backs of the mice (6.5 × 3 cm) were depilated with depilatory cream. Pictures were collected using a camera, and skin TEWL, elasticity, epidermis water content, and tissue density were measured. After data collection, the unirradiated and irradiated skin were removed, rinsed in phosphate buffer saline (PBS) on ice, and the blood was removed. The tissue block was divided into two parts: one for paraffin embedded sections, H&E staining, Masson staining, Gomori elastic fiber staining, and immunohistochemistry, and another for tissue homogenate and supernatant preparation at −80 °C, which strictly followed the corresponding biochemical kit instructions for SOD and MDA determination. Masson and Gomori elastic fiber staining were performed according to the manufacturer’s instructions provided.

After routine dewaxing and rehydration of the paraffin sections, endogenous peroxidase was eliminated with 3% H_2_O_2_ using sodium citrate solution to repair thermal antigen and washed three times with PBS for 5 min each time, enclosed in 10% goat serum for 2 h, and incubated for primary antibody at 4 °C overnight and secondary antibody for 1 h. Then, 3,3’-Diaminobenzidine (DAB) color rendering, hematoxylin counterstaining, dehydration, transparency determination, and sealing with film were performed. Images were taken with an upright microscope, and automated analysis of sample staining (percentage of positive area and positive staining cell count) was performed using ImageJ software V1.8.

Statistical analysis was performed using SPSS 20.0 software. Data were expressed as the mean ± standard deviation. Global comparisons were performed using repeated measures analysis of variance, and two independent samples *t*-tests for the same time points were considered statistically significant at *p* < 0.05.

## 5. Conclusions

In conclusion, this study designed and verified a new method to quickly establish an animal model of chronic photoaging by comparing the morphological and biological changes in C57BL/6J, ICR, and KM mice during the process of chronic photoaging. This study provides a reference for the establishment and selection of animal models of chronic photoaging in skin photoaging studies.

## Figures and Tables

**Figure 1 ijms-24-10812-f001:**
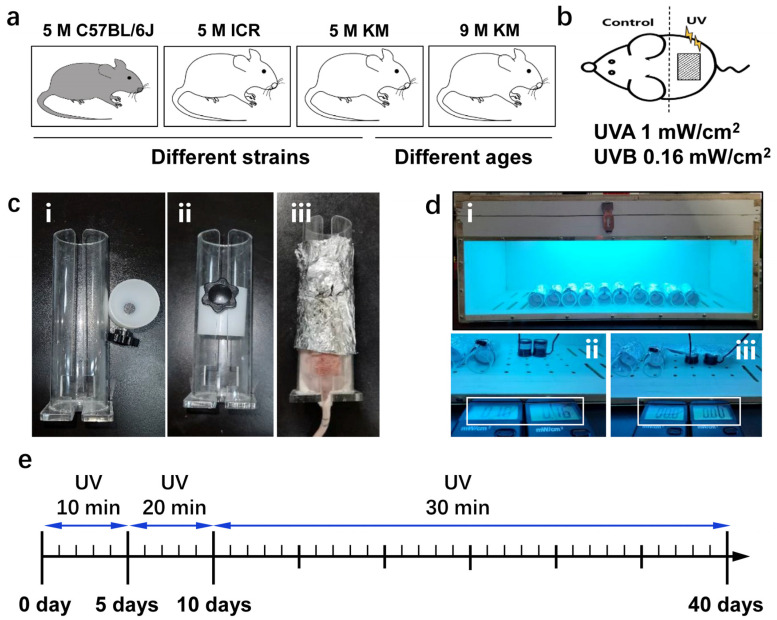
The method of chronic photoaging of the skin in mice. (**a**) Animal grouping: a 5-month-old C57BL/6J group, 5-month-old ICR group, and 5-month-old KM group were used to compare the skin changes of different strains of mice. The 5- and 9-month-old KM groups were used to compare the skin changes of mice at different ages. (**b**) UV irradiation of mice. (**c**) Mouse fixator: (i) the mouse fixator consists of the main body of the fixator and the piston. The hollowed part of the main body is used to expose the skin of the irradiated part of the mouse; (ii) overall image of the fixator; (iii) the mouse was fixed in the fixator, and its head and upper back were wrapped with silver paper. (**d**) UV irradiation box: (i) overall image of the UV irradiation box; (ii). under working conditions, the UV intensity detected by the measuring instrument is UVA 1 mW/cm^2^ and UVB 0.16 mW/cm^2^; (iii) after covering the probe with silver paper, the UV intensity detected by the measuring instrument was 0, showing that the silver paper-wrapped fixator can isolate UV and protect the head and upper back of the mice. (**e**) Time-flow chart of the chronic skin photoaging model. UV, ultraviolet; 5 M, 5-month-old; 9 M, 9-month-old.

**Figure 2 ijms-24-10812-f002:**
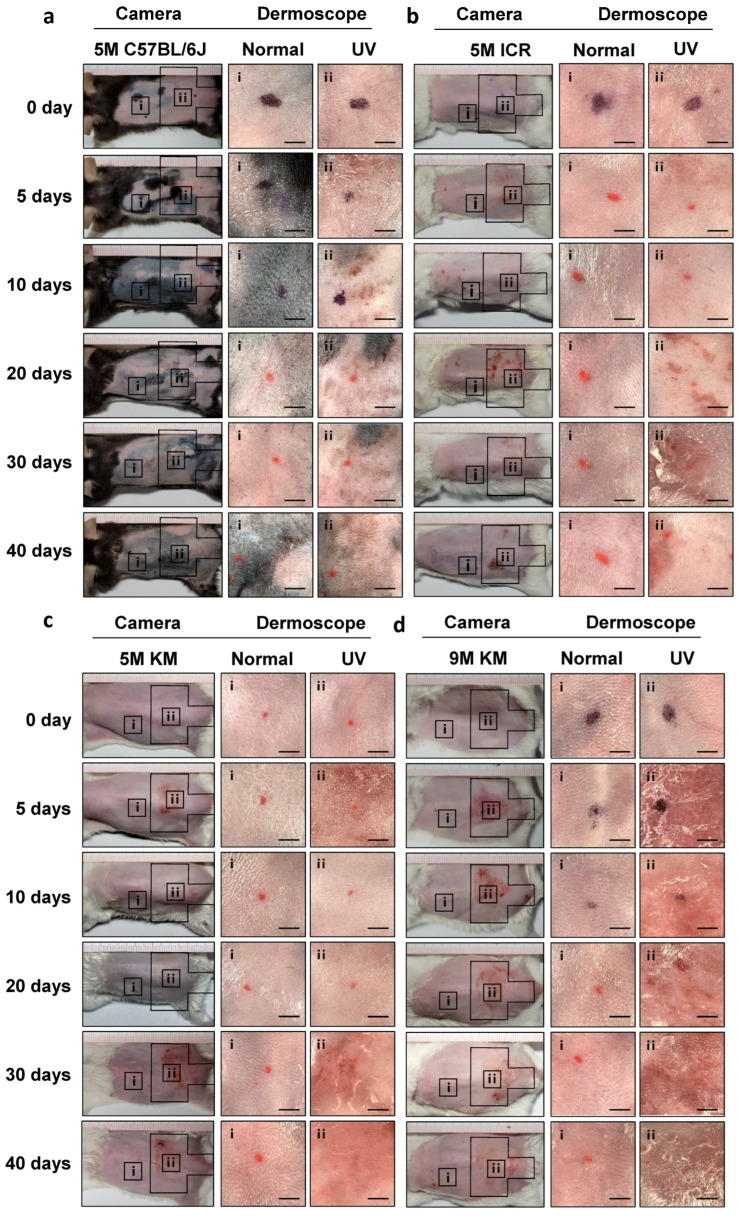
General observations of the chronic photoaging of skin in different species of mice. Photos labeled “i” show dermoscopic images of the control skin and those labeled “ii” show dermoscopic images of the chronic photoaged skin. (**a**) Dermoscopic images of 5 M C57BL/6J, (**b**) dermoscopic images of 5M ICR, (**c**) dermoscopic images of 5 M KM, and (**d**) dermoscopic images of 9 M KM. 5 M, 5-month-old; 9 M, 9-month-old; Scale bar: 25 mm.

**Figure 3 ijms-24-10812-f003:**
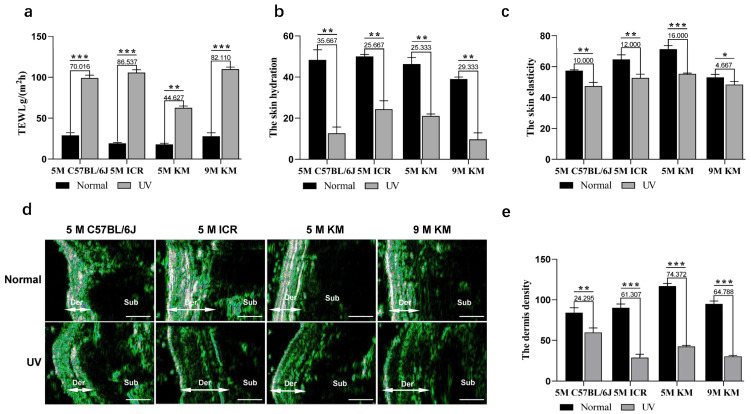
Physiological changes of the skin in chronic photoaged mice. On the 40th day, the changes in the skin physiology of the control and chronic photoaged skin of the mice were observed. (**a**) Skin barrier function was detected by percutaneous water loss (TEWL). (**b**) Epidermal hydration, (**c**) skin elasticity, and (**d**) dermal density were detected using skin ultrasound. (**e**) Quantitative statistics of dermal density were detected using skin ultrasound. The numbers above the bars indicate the specific difference between the UV irradiated sites compared to the control group. * *p* < 0.05, ** *p* < 0.01, *** *p* < 0.001. TEWL, skin percutaneous water loss; Der, dermis; Sub, subcutaneous tissue; 5 M, 5-month-old; 9 M, 9-month-old, Scale bar: 1 mm.

**Figure 4 ijms-24-10812-f004:**
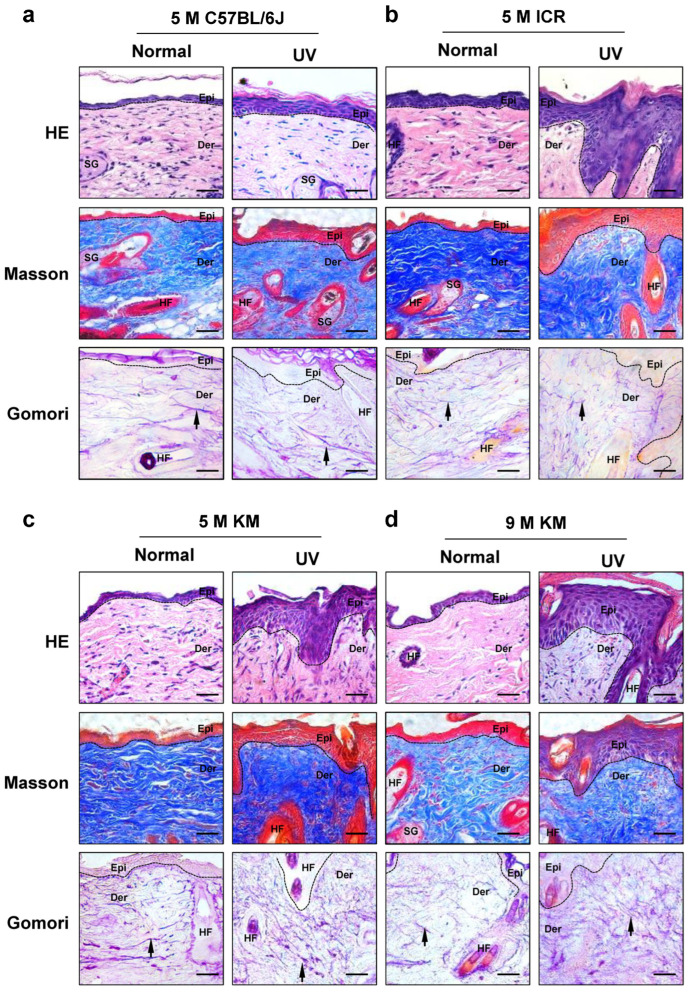
Histological changes of the skin in chronic photoaged mice. On the 40th day, the histological changes in the control and chronic photoaged skin of the mice were observed. (**a**) 5-month-old C57BL/6J mice, (**b**) 5-month-old ICR mice, (**c**) 5-month-old KM mice, and (**d**) 9-month-old KM mice. The black arrow indicates elastic fibers; HE, Hematoxylin and eosin staining; UV, ultraviolet; Epi, epidermis; Der, dermis; SG, sebaceous gland; HF, hair follicle; 5 M, 5-month-old; 9 M, 9-month-old; Scale bar: 50 μm.

**Figure 5 ijms-24-10812-f005:**
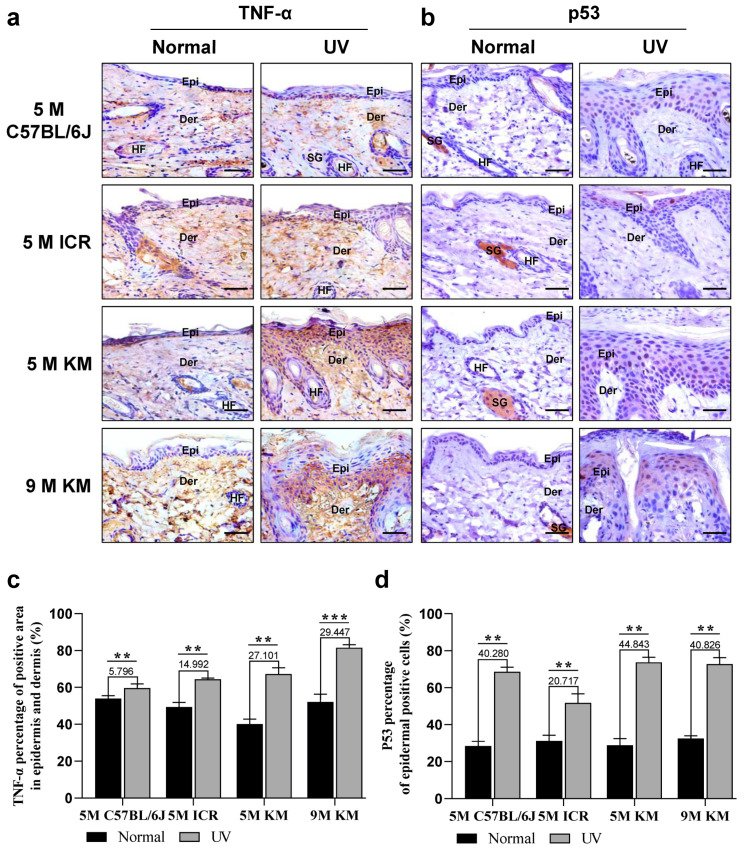
Increased expression of TNF-α and p53 in the skin of chronic photoaged mice. On the 40th day, the changes in the expression of TNF-α (**a**,**c**) and p53 (**b**,**d**) in the control and chronic photoaged skin of the mice were observed. The numbers above the bars indicate the specific difference between the UV irradiated sites compared to the control group. UV, ultraviolet; TNF, tumor necrosis factor; Epi, epidermis; Der, dermis; SG, sebaceous gland; HF, hair follicle; 5 M, 5-month-old; 9 M, 9-month-old. ** *p* < 0.05, *** *p* < 0.01; Scale bar: 50 μm.

**Figure 6 ijms-24-10812-f006:**
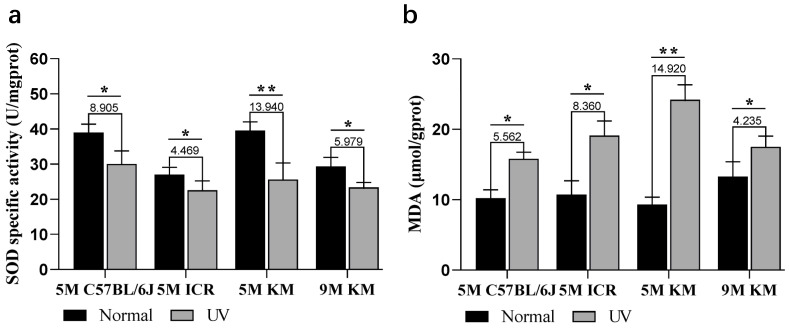
The increased skin oxidative stress of chronic photoaged mice. On the 40th day, (**a**) SOD and (**b**) MDA levels in the control skin and chronic photoaged skin were observed using ELISA. The numbers above the bars indicate the specific difference between the UV irradiated sites compared to the control group. * *p* < 0.05, ** *p* < 0.01. SOD, superoxide dismutase; MDA, malondialdehyde; ELISA, enzyme-linked immunosorbent assay; UV, ultraviolet; 5 M, 5-month-old; 9 M, 9-month-old.

## Data Availability

The authors confirm that the data supporting the findings of this study are available within the article.

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
