# Peer review of "A Comparative Study of Skin Changes in Different Species of Mice in Chronic Photoaging Models"

_ijms, 2023, doi:10.3390/ijms241310812_

Round 1

Reviewer 1 Report

UV irradiation is an exogenous mechanism of skin photoaging with numerous effects and applications e.g. in the development of novel cosmetics. Thus, skin photoaging models are very important for pharmacology as well as for cosmetics and other research purposes.

In the present study the authors have designed a new photoaging model with different mouse strains. They have analyzed the effects of UV irradiation on skin micromorphology and conducted molecular analysis. In conclusion, they found that KM is a good model for photoaging and C57BL6 for pigmentation studies. The study is interesting and important. However, I have some minor comments that must be addressed or clarified.

Specific comments

1.       What are the numbers above the bars (Figures 2, 3, 5, 6)?

2.       Why did the authors use mice 5 months old?

3.       Images should be increased. They are difficult to observe the skin architecture.

4.       Figure 5. Why did the authors only quantify the levels (staining of cells) of TNFa in epidermis? It appears that there are major differences in the dermis.

5.       What are SPF mice?

6.       In methods it is stated that 3 mice per group were used but Figure 2 shows 6 different images. How were these images obtained?

7.       When mice started the UV irradiation protocol, where they depilated? If yes how often during the 40 days of irradiation?

8.       In Figure 1c III the mouse in the fixator is depilated. Thus, it appears that depilation has taken place before irradiation.

9.       What is the central red spot in mice images (Figure 2)? Is this a wound made from depilation? This will affect the molecular analysis results.

Reviewer 2 Report

In the Introduction part, the authors should give information about the role of TNF, P53 and oxidative stress in ageing.

In the introduction, part authors should write clearly the study’s aim.

In the section Materials and Methods, the Authors should give information about the approval of the research by the appropriate ethics committee,

Moreover, the discussion part is a summary of the results, but there is no reference to other studies. In the discussion, part authors should discuss their results with the results of other authors. Now, the discussion part is insufficient.

Minor editing of English language required

Round 2

Reviewer 2 Report

The article can be published in a present form.

Minor editing of English language required